# How does a poetry audiobook app improve the perception of well-being in older adults? A study protocol

Laura Aravena-Canese[1]☯, Valeria Espejo-Videla[1]☯, Pedro O. Rossel [2,3]☯*

1 Department of Speech, Language and Hearing Sciences, Universidad de Concepción, Concepción, Chile,
2 Department of Computer Science, Universidad Católica de la Santísima Concepción, Concepción, Chile,
3 Centro de Investigación en Biodiversidad y Ambientes Sustentables (CIBAS), Universidad Católica de la Santísima Concepción, Concepción, Chile

☯ These authors contributed equally to this work.
* prossel@ucsc.cl

## Abstract

### Background

The worldwide population over 60 years of age is increasing. Thus, older adults should maintain interest and participate in social and family activities to help preserve their independence and promote their well-being. Well-being is a part of human health in its most general sense which manifests itself in all areas of human activity. Leisure activities such as listening to an audiobook may provide enjoyment and promote relaxation in older people and help improve the well-being of older adults.

### Objective

This study aims to investigate the impact of an audiobook mobile application on the well-being perception of older adults and to evaluate the usability of a mobile application specifically designed for this population.

### Methods

This protocol is a quasi-experimental study that will be conducted with 60 older adults who will use an audiobook mobile application for 4 weeks. Participants will be evaluated pre and post intervention through validated questionnaires on digital competences, usability, and perception of well-being.

### Results

A positive impact on the perception of well-being is expected in older adults who listen to poems through an audiobook mobile application.

### Conclusions

This study will allow us to know the impact on the perception of the older adult's well-being and stimulate other groups to do research on other populations and literary genres.

**Data Availability Statement:** Deidentified research data will be made publicly available when the study is completed and published. We have no data at

this time, as this article is only the protocol for the study we will be conducting.

**Funding:** This research was funded by the Dirección de Investigación of the Universidad Católica de la Santísima Concepción (Chile). POR received this fund (grant DIREG 08/2020). The funder did not and will not have any role in the study design, data collection, analysis and interpretation, decision to publish, or manuscript preparation.

**Competing interests:** I have read the journal's policy and the authors of this manuscript have the following competing interests: This research was funded by the Dirección de Investigación of the Universidad Católica de la Santísima Concepción (Chile). POR received this fund (grant DIREG 08/2020). The funder did not and will not have any role in the study design, data collection and analysis, decision to publish, or manuscript preparation.

## Trial registration

ClinicalTrials.gov Identifier—NCT05891925 (An Audiobook-based Intervention on Community-dwelling Older Adults in Concepción, Chile).

URL—https://clinicaltrials.gov/study/NCT05891925.

## Introduction

### Background

There are currently over 900 million people over 60 years of age in the world, and there will be over 2 billion by 2050 [1, 2]. In the case of Chile, it is estimated that today 18.7% of the population, i.e. approximately 3.73 million inhabitants, are more than 60 years old [3], the age that defines an older adult according to Chilean laws.

Nowadays, technology plays a crucial role in most people's lives. In general, information and communication technologies benefit older adults' quality of life by facilitating social connections, [4], reducing loneliness, aiding psychological well-being [5], and improving physical and mental health [6]. Furthermore, these technologies promote health through applications that older adults use and may help improve their quality of life perception and increase their happiness [5].

Considering the above, we will create an audiobook application to improve the well-being of older adults which can be used on their mobile phones. In addition, we will conduct a study to find out if there is an improvement in the perception of the well-being of older adults when they use this audiobook, which is a recording of a literary work read or dramatized by one or more persons to which music or effects may be incorporated [7].

The aging process generates changes in people, so it is relevant to generate comprehensive health actions to support this population through instances that contribute to their well-being and impact their quality of life [8]. Thus, an application specially designed for older adults that brings them closer to technology use by listening to poetry in an audiobook could improve their perception of well-being. According to Aguilar-Flores and Chiang-Vega [9], the use of Information and Communication Technologies (ICT) improves the quality of life of older adults given Chile's digital gap.

On the other hand, Lang *et al.* [10] indicate that the use of audiobooks contributed positively to the quality of life and well-being of older adults with vision loss. Best et al. [11] mention that listening to audiobooks keeps adults' brains stimulated, makes them feel less stressed and less anxious, more animated, and gives them a feeling of companionship as long as they have devices to listen to them or can afford them. In addition, reading poetry aloud is favorable for older adults [12], but there is a need for more rigorous studies that can demonstrate improvements in their health and well-being.

According to Lee *et al.* [13], well-being refers to individuals' affective and cognitive evaluations of their lives, and is associated with key factors such as income, health, basic needs fulfillment, relationships with others, and demographic (age and gender) factors.

Even though there is not a total agreement on what well-being is, [14] Ryan and Deci [15] have indicated that well-being can be organized into two big traditions: the hedonic (subjective) view and the eudaimonic (psychological) view. The first view relates to happiness and how the goal of life is to experience the maximum amount of pleasure, and is broadly focused on wishes and self-interests. The second view is linked to the development of human potential

and the realization of what one can become. The term eudaimonia is invaluable as it refers to well-being as distinct from happiness per se.

Apart from those two types of well-being, there is also social well-being to consider. This is, in simple words, the assessments that people make regarding their circumstances and functioning within society, and is composed by the following dimensions: [16] social integration, social acceptance, social contribution, social actualization, and social coherence.

Most older adult population is not using many information technologies and mobile applications, due to physical, acceptance and design barriers [17]. Pang *et al.* [18] described that older people felt a lack of confidence and vacillation, because they felt that they learn more slowly and tend to pose the same questions repeatedly to technical support. However, there are also facilitators in the use of technology focusing on e-health by the older adult, such as motivation, the desire to learn, the sense of curiosity, the benefits for health, among others [19].

Thus, the problems faced by older adults using technology block them from using services that could have a positive impact on their health, such as cultural content, books, stories or poems that are accessible on virtual platforms.

According to Deshpande [20], the advantages of reading poetry are that it allows people, on the one hand, to express emotions that they would normally have hidden and, on the other hand, to express their real feelings about simple or everyday things.

In the work of Healey et al. [12] with older adults, the analysis of group conversations coupled with literary analysis concluded that poetry has the potential to act as a catalyst for discussion and self-disclosure, which can have beneficial effects on people's well-being.

Audiobooks have some advantages over traditional books because listening, rather than watching, can evoke a more emotional response to the story and achieve greater emotional and psychological engagement with the text [21]. Furthermore, a study done by Cheng and Wang [22] showed that audiobooks are a valuable tool for promoting mental health and, thus, well-being in older adults, especially those with limited reading capabilities.

Finally, to the best of our knowledge, no studies relate the well-being of older adults to listening to poetry on an audiobook.

## Objective, research question and hypothesis

The goal of this 4-week quasi-experimental study is to assess the impact of an audiobook mobile application on the well-being perception of older adults belonging to a Community Rehabilitation Center (CRC) at Concepcion, Chile, and to evaluate the usability of the mobile application designed for this population.

Our research question is the following: Does listening to audio poems using a mobile application improve the perception of well-being of the older adults who go to the CRC at Concepción?

Finally, our hypothesis is that using a mobile application that allows listening to poems produces a positive change in older adults' perception of well-being.

## Materials and methods

### Mobile application description

Our audiobook application was designed to be simple, as it was designed and built specifically for this study. It was assessed following the older adult heuristics from Silva *et al.* [23] for this population. It is designed to run on smartphones running Android 5.0 or higher.

The application has three main functions, organized into a footer app navigation menu with three elements: Library, List, and Player. Furthermore, there is also a typical user login screen.

In the following paragraphs, we will explain the application functionality through mockups. The information at the top is consistently presented in all figures and consists of the name of the application and a three points menu.

The first element (**Library**) is shown in Fig 1. When in use, the footer app navigation icon **Library** is highlighted and the other two icons are faded. In this mode, the application presents the audiobooks available to the older adults. Several of them can be loaded in the application.

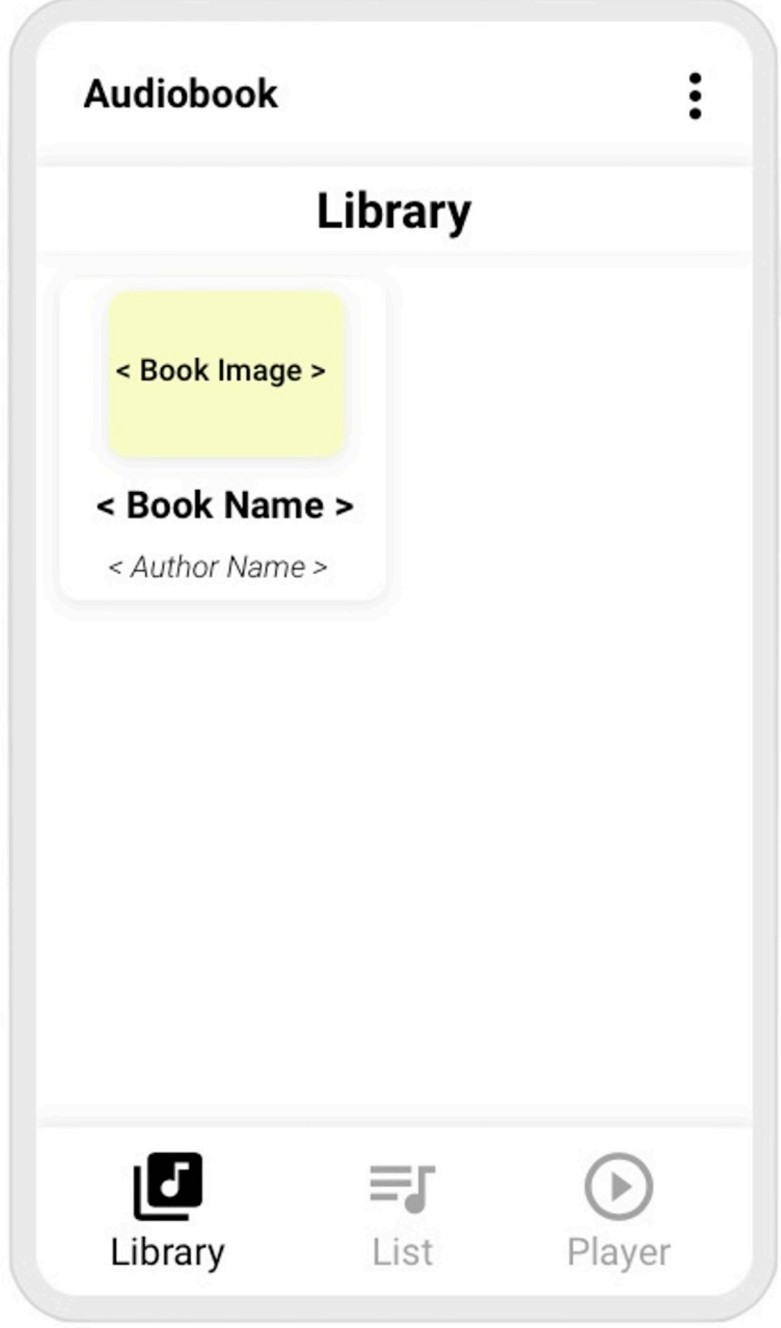

**Fig 1. Audiobook mockup, library.**

Each book is represented by a square with rounded corners, containing the book's name, its author's name and an image of the book's cover.

When the user selects the book by pushing its icon, the next element (**List**) is displayed in the smartphone. This is shown in Fig 2. Similarly, the footer app navigation icon **List** is highlighted and the other two are faded. In **List**, the book name is shown on top, and below it a search icon with its corresponding input field is shown, through which a user may find an

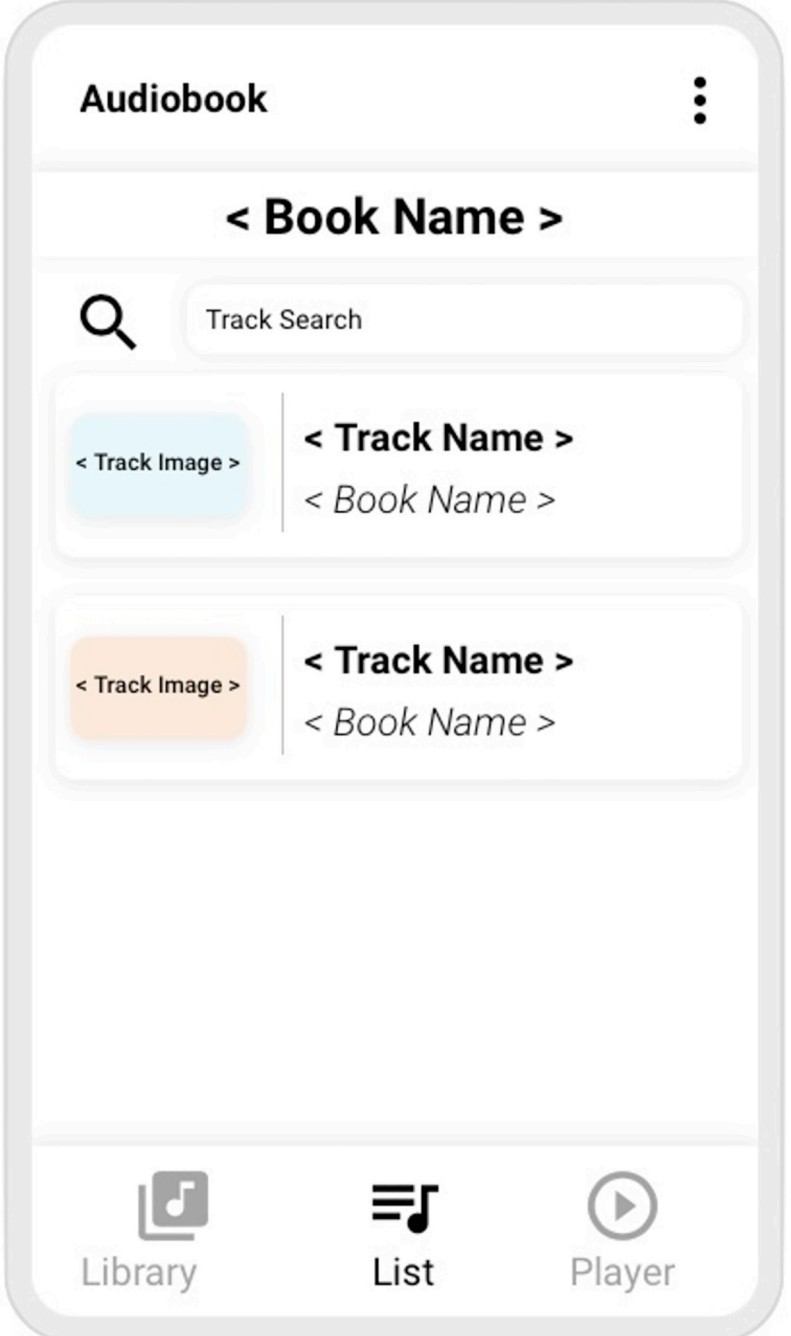

**Fig 2. Audiobook mockup, list.**

specific track from the audiobook. Furthermore, it is useful as a filter: as text is entered, the tracks that match it appear below the icon.

Tracks are shown as a rectangle with rounded corners containing an image of the track, its name and the name of the book to which it belongs.

When the older adult selects the track by pushing its icon, the next element (**Player**) is activated in the smartphone, as shown in Fig 3. Again, the footer app navigation icon **Player** is

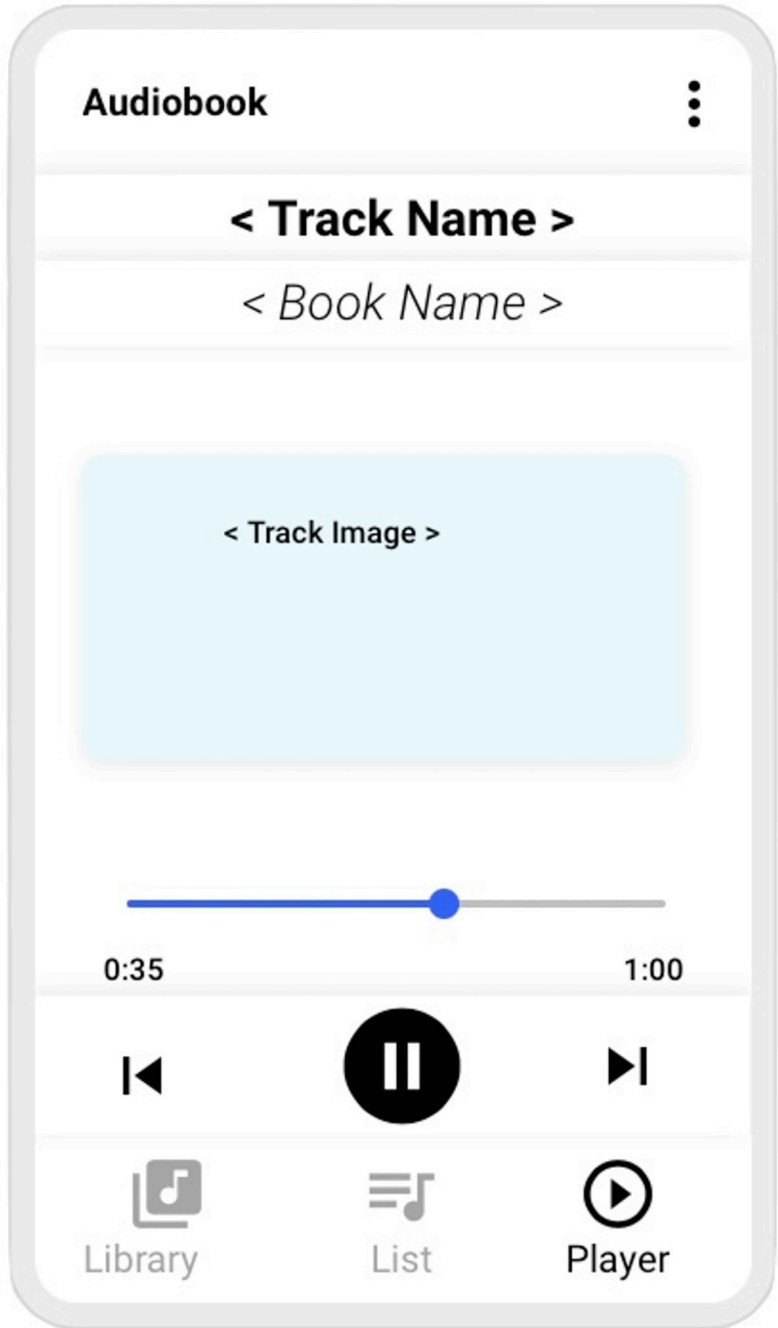

**Fig 3. Audiobook mockup, player.**

highlighted and the other two are faded. Now, the track name and book name are shown at the top. An image related to the track is centered a few lines below. Furthermore, the track's duration in minutes and seconds is shown on the right side, along with a blue slider ribbon representing the current position along the track. Finally, three icons are displayed on the navigation footer to control track reproduction: the Skip Previous button (on the left), the circular Pause or Play button (in the center), and the Skip Next button (on the right).

In general, a track may be a poem, a book chapter, or any other structure into which an audiobook can be divided. In this case, each track represents a poem from the book "Ternura" by Gabriela Mistral, who was a Chilean poet and educator recognized as one of the most important and influential literary figures in Latin America, particularly in the field of poetry, receiving the Nobel Prize in Literature in 1945 [24].

This application version will gather usage data, which will be stored both in a local database and a remote one: date, hour and reproduction time of the track (poem), name of the poem, book to which the poem belongs, and older adult id.

This data will be stored locally until the application can communicate with the remote database, at which point it will store the data remotely and then delete it from the local store.

The remote database will be stored on a paid server, which guarantees the privacy, accuracy, and availability of the stored data, as well as avoiding any damage or misuse by providing protection against unauthorized access and data leaks. Access to this database will be allowed only to one of the researchers (POR), who can download the information for analysis. Afterward, this information will be deleted from the server.

## Design

A quasi-experimental study will be conducted during the second quarter of 2024 to assess the improvement in the well-being perception of older adults and the usability of an audiobook mobile application. This design is based on the desire to reduce ethical considerations (ethical problems of randomizing patients), avoid data contamination, and offer a more cost-effective and feasible approach for implementing this research [25–27]. In this study, participants will be compared with themselves at the beginning and at the end of the intervention, that is, a pretest/posttest with a single group.

Fig 4 shows the study protocol's flow diagram. The first stage at the top corresponds to recruiting community-dwelling older adults, to be done at a community center (recruitment will begin at the end of April 2024 and be completed at the end of May 2024) by two of the researchers (LA-C, VE-V). With the older adults who agree to participate in the study, we (LA-C, VE-V) will do the second stage (screening), which consists of determining who can participate in the study by applying the inclusion and exclusion criteria (third stage). Before starting the intervention, participating older adults must accept and sign a written consent (fourth stage). Some of them may refuse to participate in the study and thus be excluded from it (fifth stage). Then, a baseline (sixth stage) will be established by applying standardized and validated tests to the participating older adults, who will then have an induction session, in which the three researchers will participate (LA-C, VE-V, POR), to install the application on their smartphones, learn how to use the application and get answers to any questions they might have (seventh stage). In the intervention (eighth stage), the participants will use the application for four weeks (during June 2024), with a follow-up at 2 weeks (ninth stage) to be performed by two of the researchers (LA-C, VE-V). After these four weeks of application use, the older adults will be evaluated through standardized tests (tenth stage). Finally, the data obtained from the questionnaires and from using the audiobook itself will be analyzed (eleventh stage).

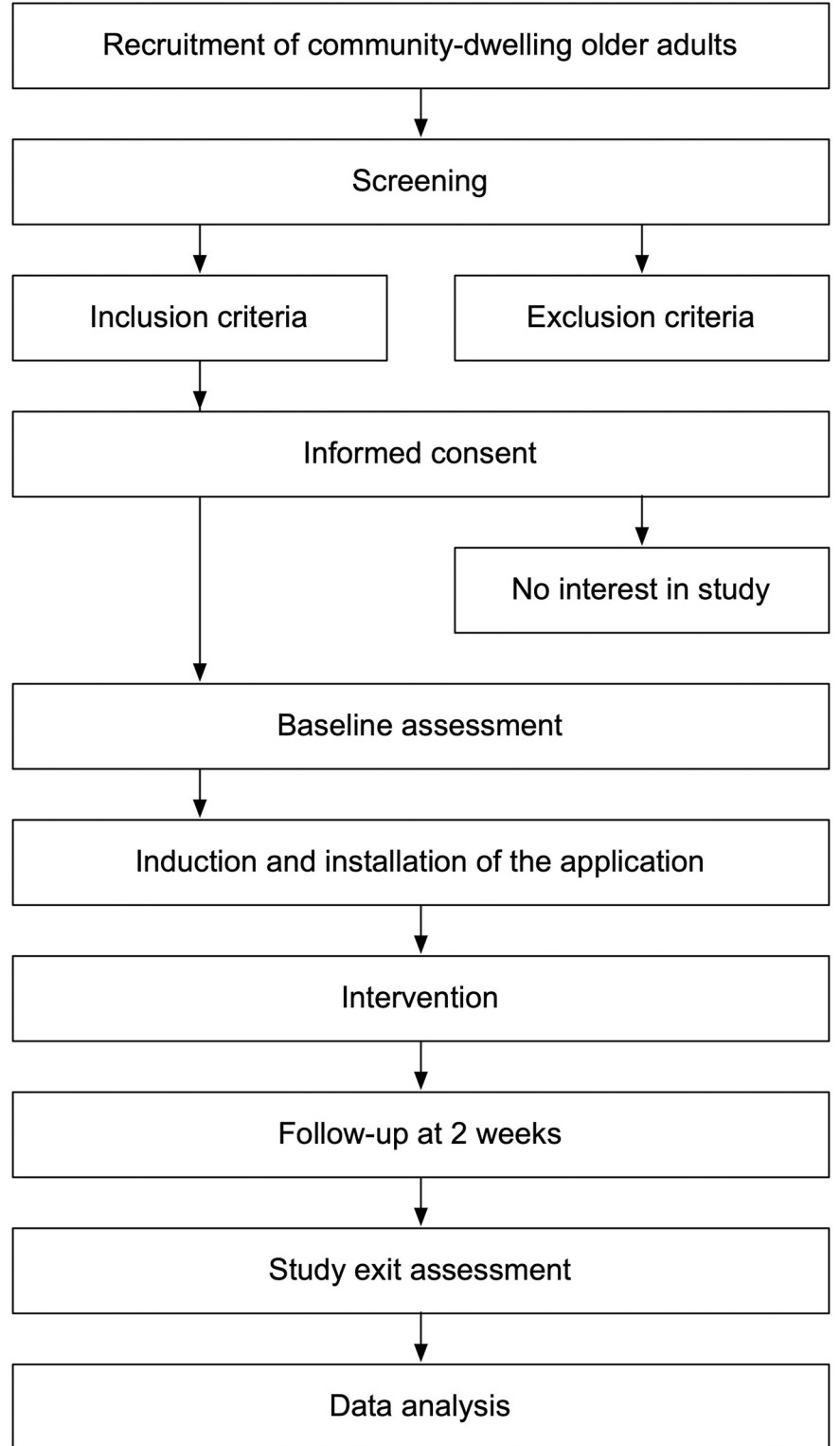

**Fig 4. Study design overview.**

In the sections Recruitment and Intervention, we will discuss the above mentioned stages in detail. Furthermore, a Standard Protocol Items Recommendations for Interventional Trials (SPIRIT) checklist is provided as S1 Checklist.

## Recruitment

The participants will be recruited from the Community Rehabilitation Center at Concepción, Chile. We estimated our sample size considering a pre-post difference of 1,24 points on the mental health outcome [28] and a potential standard deviation of 3,4. Thus, we planned for a sample size of 60 individuals [29] with a continuous response variable measured pre-post intervention to be able to reject the null hypothesis that the response difference is zero with power 0,8 and an error probability of 0,05. The sampling technique will be non-probabilistic and will be used for the convenience of the study.

Recruitment flyers will be posted on the walls of the CRC to make an open call to all people over 60 years of age being treated at that place. Furthermore, group meetings scheduled by the researchers (LA-C, VE-V) will be held at the CRC offices to invite older adults to participate in the research. Older adults interested in the study will register by giving their name and telephone number, and will be contacted later by the researchers (LA-C, VE-V).

Older adults interested in participating in the study will be contacted by telephone. Then, the study, the objectives of the research and the strategy to be used in the study will be explained to them in a concise and precise way. Furthermore, they will be questioned to find out if they meet the inclusion criteria for the study. Inclusion and exclusion criteria are shown in Table 1.

If the older adults refuse to participate, they will be thanked and told there is no harm to them. Otherwise, they will be invited to a meeting at the CRC, on a schedule to be agreed.

During the meeting, the written informed consent will be handed out and explained to them with the support of the consent itself, emphasizing that participation in the study is voluntary, and that the confidentiality of the each participant's personal information will be safeguarded. If the older adult agrees to the consent, it will be signed by them and by the researcher responsible for the research (VE-V), and the participant will receive a signed copy of the consent.

Then, two questionnaires will be applied to the older adults: one to measure their digital skills with a validated Spanish questionnaire based on DIGCOMP [30, 31], and another to assess their well-being prior to the experience with the audiobook application (Part A). This last questionnaire was built by the authors specifically for this application (see S1 Appendix for details). It consists of two parts, and both the content and face validity were made through by

**Table 1. Inclusion and exclusion criteria.**

| Inclusion Criteria | Exclusion Criteria |
|---|---|
| • Person aged 60 and over who belongs to the CRC at Concepción.<br>• Speak and read Spanish.<br>• Enabled and capable of approving the informed consent.<br>• Provide signed informed consent.<br>• Older adults who have a smartphone where they can run the audiobook mobile application.<br>• Older adults who are not participating in other emotional, social and/or motor training programs. | • Significant psychiatric or medical illness (depressive disorder, delirium, intellectual disability, etc).<br>• Older adults classified with dependency criteria.<br>• Illiterate people.<br>• Presence of a severe sensory deficit, either in visual or hearing difficulties.<br>• Presence of neurodegenerative diseases (dementia, Alzheimer's disease, Parkinson's disease, multiple sclerosis, etc.). |

an expert committee, with clinical experience, knowledge in methodological investigation, and disciplinary knowledge in psychology [32].

Finally, one of the researchers (LA-C) will explain to the older adults how the application works, and answer any questions they may have. After the explanation, the audiobook application will be installed on the older adult's smartphone, and they will be provided with a contact telephone number to answer any questions about the operation of the application.

## Intervention

The intervention will consist of the use of the audiobook application for 4 weeks. During this period, the application will record the following data autonomously:

- older adult id,

- date, hour and time that the older adult listened to each poem,

- poems listened.

This data will be stored locally and sent to a server for remote storage when the older adult has internet access.

Considering the hypothesis, there is no minimum application frequency of use to be suggested to participants. Thus, older adults should use the audiobook application according to their time availability and their motivation. Furthermore, participants are advised to abstain from any other emotional, social and/or motor training during the intervention period.

After two weeks of application use, the researchers (LA-C, VE-V, POR) will carry out a follow-up in order to answer any questions that may arise from the use of the application, and explain again the objectives of the research, if necessary. This follow-up will be initially done by telephone, and in person if required.

After four weeks, the older adult will be invited to a meeting at the CRC, on a schedule to be agreed beforehand, to answer the well-being questionnaire (Part B) and a validated Spanish usability questionnaire (SUS [33, 34]).

Finally, data analysis will be carried out once the data gathering is finished.

## Outcome measures

The following variables will be recorded:

- Socio-demographic and digital competences variables

  - Age: number of years since the older adult was born until the day of the interview.

  - Gender: man or women according to own definition.

  - Education level: number of years of formal schooling completed by the older adult.

  - Digital competences: set of knowledge, skills, and attitudes required when using Information and Communication Technologies and digital media to perform tasks, solve problems, communicate and manage information, and build knowledge efficiently, appropriately, and autonomously. It will be measured through a questionnaire based on DIGCOMP [31]. In it, five areas of relevant competences are identified: information and data search, communication and collaboration, digital content creation, safety, and problem solving. The questionnaire allows to determine the older adults' competences. The way to categorize the digital competences of each participant is described in the same work by García Díaz and Villafañe [31].

- Dependent variables. These variables are related to well-being perception, and will be measured through a questionnaire specifically created by the authors to know the well-being of the older adults before (Part A) and after (Part B) the use of the audiobook application. Part A includes 6 questions and Part B includes 12 questions that are assessed with a 5-point Likert subjective scale (1 strongly disagree and 5 strongly agree). These questionnaires measured the following types of well-being: Hedonic, Eudaimonic, and Social.

- Independent variables.

  - Usability: It is the ease with which users can use software. It will be measured through the System Usability Scale (SUS) questionnaire [34]. It consists of 10 questions that are assessed with a 5-point Likert subjective scale (1 strongly disagree and 5 strongly agree). The result is between 0 and 100; the higher the result, the higher the usability level of the application. The overall score calculation method is detailed in the work of Sevilla-Gonzalez et al. [34], and the results interpretation can be found in Bangor et al.'s research [35].

  - Frequency of use of the application: Number of poems listened to per day.

## Data collection procedure

Once participants are contacted, informed and agree to participate in the study, data will be collected for subsequent statistical analysis. Some information will be recorded using paper and pencil. The documents will be applied by themselves, and in this instance one of the researchers (VE-V) will monitor the participants to answer any questions they may have. Others will be recorded through the audiobook application.

Data security and confidentiality will be safeguarded throughout the complete information collection process.

Sociodemographic data will be individually asked to each older adult during the baseline assessment. Specifically, we will obtain information about age, gender, and educational level.

Secondly, we will apply the DIGCOMP questionnaire to assess the older adults' digital competence. Furthermore, part A of the well-being questionnaire will be applied obtaining the current Hedonic, Eudaimonic, and Social well-being perception before using the application.

The duration of the intervention will be 4 weeks. During this period, the audiobook application will collect information automatically every time the audiobook application is used. Information to be collected include the poems listened, date, hour and reproduction time that the participants listened to each poem. This information will be stored in a remote database.

In the exit assessment, the application of part B of the well-being questionnaire will be done, obtaining the perception of the older adults' Hedonic, Eudaimonic and Social well-being after the intervention. In addition, the usability questionnaire (SUS) will be applied to the older adults to assess the audiobook application's usability.

## Statistical analysis

For data analysis, a single database will be created with the results obtained from the application of the questionnaires, the older adults' sociodemographic characterization, and the information collected automatically by the application.

Subsequently, descriptive statistics will be used for univariate and bivariate frequency analysis which will consider measures of central tendency (mean and standard deviation) to describe the characteristics of the sample and the questionnaire results.

Then, the study hypothesis will be carried out using a multivariate linear regression in order to characterize the well-being variable at the end of the intervention, controlled by the initial well-being, and by sociodemographic variables, usability and frequency of use.

We will explore if the covariates play a role of confounding effects and/or modifiers in the relationship between the initial and the final well-being score. The specific model is assumed to be a Generalized Linear Model, for normal distribution and identity link. Data processing and analysis will be performed using the IBM SPSS Statistics for Windows, Version 28.0 (IBM Corp. Released 2021, Armonk, NY, USA).

Data analysis will be performed under an intention-to-treat (ITT) approach by a researcher blinded to the study group. The multiple imputation technique [36] will be applied using the predictive mean matching method to handle missing data and ensure the inclusion of all participants in the analysis.

The proposed statistical model is summarized as follows:

$$B_{final} = a_0 + b(B_{initial}) + b(frec) + b(usability) + b(gender) + b(other\ dem)$$

where

$B_{final}$ : Final well-being score.

$a_0$ : Intercept.

$b(B_{initial})$ : Slope associated with initial well-being.

$b(frec)$ : Slope associated with the application's frequency of use.

$b(usability)$ : Slope associated with the application's evaluated usability.

$b(gender)$ : Slope associated with the participants' gender.

$b(other\ dem)$ : Slopes associated with other confounding demographic variables or relevant modifying effects.

## Limitations of the study

We must consider that some older adults' smartphones may be stolen, lost or damaged. This study will register any smartphone losses, and in this case, we will ask the older adult if they can continue in the study with another smartphone or if they will have to leave the study. In the last case, we will have to recruit another older adult meeting the inclusion criteria.

The smartphone may run out of battery during the listening of the poems. To minimize this inconvenience, older adults will be warned about this possibility in the "Induction and installation of the application" step (see Fig 4), and they will be encouraged to ensure that smartphones are always charged during the use of the audiobook application.

The literary genre chosen on this occasion (poetry) might not be the one preferred by most older adults participating in the study. At the end of the study, we should ask them and introduce other genres in another version of the audiobook application.

The resurgence of COVID-19 in Chile could make it necessary to extend the recruitment period for older adults, and also increase the total time budgeted for the study.

Finally and considering the above, older adults could be reluctant to participate in the study or meet with researchers in person, in which case we will opt for an online modality, either during recruitment, during follow-up, or when answering the questionnaires.

### Ethical considerations

The Scientific Ethics Committee of the Concepción Health Service in Concepción, Chile, approved this study on May 13th, 2021 (code CEC-SSC:21–01-06).

The researchers declare compliance with the Good Clinical Practices established by the Government of Chile. Furthermore, the research will ensure compliance with Law 20,584 regarding the Rights and Duties of patients (Ley 20,584 de Derechos y Deberes de los pacientes, in Spanish) and the ethical norms of the Declaration of Helsinki.

Participation of older adults in this research will be voluntary: they may withdraw at any chosen time, without any harm to them. The above will be stated in the informed consent form, which will be first presented verbally to the participants. Then, they will be asked to voluntarily sign a written copy, having previously received all relevant information about the process and objectives of the study.

Participants will not receive any direct benefit, reward, or economic compensation of any kind for participating in this study. Furthermore, the results will be made available to them.

The research team will protect each participant's information, privacy and confidentiality. Data obtained will only be used to fulfill the objectives of this study. One of the researchers (VE-V) will be in charge of safeguarding both the data obtained through the application and those obtained from the applied questionnaires. Once the research has finished, the data will be stored on a USB storage device for two years. It will be safely kept in a locked drawer in the office of the responsible researcher (VE-V), who has sole access to it in the Department of Speech, Language, and Hearing Sciences, Universidad de Concepción, Chile.

## Results

As mentioned, recruitment will start at the end of April 2024, and we expect to enroll an estimated number of 60 older adults. Recruitment should last approximately one month, and a follow-up will be done in June 2024. Finally, the data collection should be complete by July 2024.

If the results confirm our hypothesis, i.e., that listening to audio of poems using a mobile application improves the perception of well-being of older adults, we could test the effectiveness of this intervention considering other aspects: (1) adding other literary genres (or subgenres) to the audiobook such as drama, fiction and nonfiction; (2) incorporating a greater number of books for each genre; (3) carrying out the intervention with another population, such as people in another age range, or blind people, among others.

## Discussion

This paper provides a study protocol to evaluate the impact of an audiobook mobile application on the perception of well-being of older adults. We are interested in studying three types of well-being (Hedonic, Eudaimonic, and Social). Furthermore, we will collect data with the help of the audiobook mobile application and through validated questionnaires.

Considering that the application will run over Android 5.0 or higher for smartphones, we postulate that the intervention will not have problems related to the installation and subsequent execution of the application in several different models and brands of phones.

Poerio and Totterdell [37] found a positive effect on various types of older adults' well-being (Hedonic, Eudaimonic, and Social) and on their sense of meaning in life when using audiobooks of the fiction and nonfiction genres. According to the above, there is the option of investigating other literary genres and their impact on older adults' well-being. In particular, it may be interesting to investigate how well-being improves according to the choices in audiobook literary genres made by older adults [28].

On the other hand, technical problems or difficulties may arise when older adults use the audiobook application. To overcome those issues, the authors will be available to provide technical support by email, phone or face-to-face if needed. The above could affect the adherence to the intervention. In order to prevent low adherence, the audiobook application was assessed following the older adult heuristics specific for this population from Silva *et al.* [23]. If the adherence problem occurs despite the above, other methods or heuristics should be used in future works.

As future work, we may do an intervention with a larger sample size, with other populations such as adults, community-dwelling older adults, or older adults in long-term care facilities, and also to carry out the intervention with a qualitative approach considering focus groups, among others. For example, people with Alzheimer's disease reported feelings of discomfort and some aversion related to the audiobook experience, although the population was small [38]. This offers the possibility of analyzing other populations such as older adults with cognitive impairments.

## Conclusions

We have presented a study protocol to research how a poetry-based audiobook application for smartphones impacts the well-being perception of older adults. In particular, this study is the first to explore the effects of an audiobook in the well-being of the older adult population in the CRC at Concepción, Chile.

From a research point of view, this study can motivate and guide other research groups in doing research on the well-being of other populations and thus gather new data and knowledge about their well-being.

## Supporting information

**S1 Checklist. SPIRIT 2013 checklist.** Recommended items to address in a clinical trial protocol and related documents.
(PDF)

**S1 Appendix. Well-being questionnaire.** A questionnaire to obtain the older adults' perception of Hedonic, Eudaimonic and Social well-being before and after the intervention.
(PDF)

**S1 File.**
(PDF)

**S2 File.**
(PDF)

## Acknowledgments

The authors thank Patricia A. Huerta and Patricia Pérez-Wilson from Universidad de Concepción for providing help with the study's statistical analysis and the well-being questionnaire's construction. Furthermore, we thank Antonio Bascur and Pamela Salamanca-Flores for providing support in the audiobook mobile application's design, and David Aguayo-Vidal for encouraging the development of this research. Finally, we thank Proyecto Ingeniería 2030 (ING222010004) from Universidad Católica de la Santísima Concepción, Chile.

## Author Contributions

**Conceptualization:** Laura Aravena-Canese, Valeria Espejo-Videla, Pedro O. Rossel.

**Investigation:** Laura Aravena-Canese, Valeria Espejo-Videla, Pedro O. Rossel.

**Methodology:** Laura Aravena-Canese, Valeria Espejo-Videla, Pedro O. Rossel.

**Project administration:** Valeria Espejo-Videla.

**Visualization:** Pedro O. Rossel.

**Writing – original draft:** Laura Aravena-Canese, Valeria Espejo-Videla, Pedro O. Rossel.

**Writing – review & editing:** Laura Aravena-Canese, Valeria Espejo-Videla, Pedro O. Rossel.

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
