## [Decision Letter · Decision Letter 0]

25 Jun 2024

PONE-D-24-11113How Does a Poetry Audiobook App Improve the Perception of Well-being in Older Adults? A Study ProtocolPLOS ONE

Dear Dr. Rossel,

Thank you for submitting your manuscript to PLOS ONE. After careful consideration, we feel that it has merit but does not fully meet PLOS ONE’s publication criteria as it currently stands. Therefore, we invite you to submit a revised version of the manuscript that addresses the points raised during the review process.

We look forward to receiving your revised manuscript.

Kind regards,

Junchen Shang

Academic Editor

PLOS ONE

Journal Requirements:

3. Please remove your figures from within your manuscript file, leaving only the individual TIFF/EPS image files, uploaded separately. These will be automatically included in the reviewers’ PDF.

4. We note that the original protocol file you uploaded contains a confidentiality notice indicating that the protocol may not be shared publicly or be published. Please note, however, that the PLOS Editorial Policy requires that the original protocol be published alongside your manuscript in the event of acceptance. Please note that should your paper be accepted, all content including the protocol will be published under the Creative Commons Attribution (CC BY) 4.0 license, which means that it will be freely available online, and any third party is permitted to access, download, copy, distribute, and use these materials in any way, even commercially, with proper attribution.

Therefore, we ask that you please seek permission from the study sponsor or body imposing the restriction on sharing this document to publish this protocol under CC BY 4.0 if your work is accepted. We kindly ask that you upload a formal statement signed by an institutional representative clarifying whether you will be able to comply with this policy. Additionally, please upload a clean copy of the protocol with the confidentiality notice (and any copyrighted institutional logos or signatures) removed.

Reviewers' comments:

Reviewer's Responses to Questions

**Comments to the Author**

1. Does the manuscript provide a valid rationale for the proposed study, with clearly identified and justified research questions?

Reviewer #1: Yes

Reviewer #2: Yes

2. Is the protocol technically sound and planned in a manner that will lead to a meaningful outcome and allow testing the stated hypotheses?

Reviewer #1: Partly

Reviewer #2: Yes

3. Is the methodology feasible and described in sufficient detail to allow the work to be replicable?

Reviewer #1: No

Reviewer #2: Yes

4. Have the authors described where all data underlying the findings will be made available when the study is complete?

Reviewer #1: Yes

Reviewer #2: No

5. Is the manuscript presented in an intelligible fashion and written in standard English?

Reviewer #1: Yes

Reviewer #2: Yes

6. Review Comments to the Author

You may also provide optional suggestions and comments to authors that they might find helpful in planning their study.

Reviewer #1: Comments

Figure 3 needs to be revised to improve its flow and alignment with the information presented on Page 6. Additional details are to be incorporated into the figure.

There was no list of references attached to this manuscript. There were errors in the presentation of the cited reference in the PDF version e.g The cited references in the text indicated [?}.

Line 91, the statement ‘The goal of this protocol is to describe a quasi-experimental study’ to be revised.

Line 94-95, the statement ‘Furthermore, this study will evaluate the usability of the mobile application designed for this population. The duration of the study will be 4 weeks.’ is to be written in the form of an objective and in one sentence.

More information on the data storage/server is to be provided.

Line 170-175, proper sample size calculation is to be provided, before and after the inclusion of attrition rates. Sample size formula or software used to derive the sample size is to be stated.

In Page 7 Line 185, it was written Figure 4 Study Design but in the attachment is was written Figure 3.

Quasi-experimental designs typically requires a comparison group. The reason to conduct one-arm design is to be stated. Although some information about changes in the outcome over time within the same group could still be obtained, it is difficult to determine if the observed changes in the outcome are actually due to the intervention being studied. Also, it is more susceptible to threats of internal validity.

Line 197, the information on the validity of DIGCOMP in the context of the local population is to be provided. Likewise with the System Usability Scale. Ensure all questionnaire validation information is described where applicable. The scoring method is to be provided if any.

Description on missing data (if any) and handling method is to be provided.

The researcher abbreviation name involved in the study could be added to identify the role.

Line 348-349, for this statement ‘The data will be stored in a USB storage device for two years once the research has finished’ how the authors ensure the security of this device is to be stated.

Reviewer #2: It makes sense to focus on how to improve the well-being of older adults. It is interesting to describe a quasiexperimental study to assess the impact of an audiobook mobile application on the well-being perception of older adults. However, there are many issues in the paper that need to be improved.

1.The logic of the introduction section needs to be strengthened, the background in the introduction is a bit lengthy. The motivation for the initiation of this research program is currently unclear. As audiobook apps are already commonplace in many countries, the article needs to emphasize the necessity and innovativeness of the design of the study protocol. It is therefore recommended that the introduction part of the paper need to be reorganized.

2.The standardization of the format of the paper needs to be strengthened, with citations and references missing from the paper.

3.Is the sample size set in the methodology section reasonable? Please describe the rationale for sample size selection.

4.P353:“recruitment will start at the end of April 2024, and we expect to enroll between 30 and 60 older adults.” It mentioned a focus on the well-being of 60 year olds in the background, why was recruitment targeted at the 30-60 age group?

7. PLOS authors have the option to publish the peer review history of their article (what does this mean?). If published, this will include your full peer review and any attached files.

Reviewer #1: No

Reviewer #2: No

---

## [Author Response · Author response to Decision Letter 0]

9 Sep 2024

Dear Editor and Reviewers.

The authors want to thank you for your thoughtful and constructive comments. Our manuscript has been substantially improved thanks to your observations. Below, you will find an itemized point-by-point response to the reviewers’ comments.

General Comments

Comment 2. Is the protocol technically sound and planned in a manner that will lead to a meaningful outcome and allow testing the stated hypotheses?

Reviewer #1: Partly

Reviewer #2: Yes

Response 2. We appreciate the comment. The “Materials and methods” section has been improved by taking into account the suggestions made by the reviewers. We have modified all its subsections so as to strengthen the final document. For more details, please review the changes made to the document itself. 

Comment 3. Is the methodology feasible and described in sufficient detail to allow the work to be replicable?

Reviewer #1: No

Reviewer #2: Yes

Response 3. As in the previous comment, we have improved all subsections of the “Materials and methods” section, taking into account the reviewers’ suggestions.

Comment 4. Have the authors described where all data underlying the findings will be made available when the study is complete?

Reviewer #1: Yes

Reviewer #2: No

Response 4. We reviewed the protocol approved by the Scientific Ethics Committee which was previously uploaded. We mistakenly interpreted the requirements established in the journal form when uploading the article for review. That protocol states that the participants’ information will not be shared; however, the procedures, intermediate data, and results can be freely published, as required by the PLOS Editorial Policy, provided that personal data are properly anonymized.

On the other hand, we have no data at this time, as this article is only the protocol for the study we will be conducting. However, all data underlying the findings of this study will be made available upon request or as supplementary material once the study is completed. The data will be provided in a standard format and under a specified open license type depending on the journal in which the results are published. The data will be completely anonymized to protect the participants’ privacy. 

Reviewers Comments

Reviewer 1

Comment 1. Figure 3 needs to be revised to improve its flow and alignment with the information presented on Page 6. Additional details are to be incorporated into the figure.

Response 1: In figure 3, we replace the following paragraph: 

"Furthermore, the track's duration in minutes and seconds is shown, along with a blue slider ribbon representing the current position along the track. Finally, three icons are displayed on the navigation footer to control track reproduction: skip previous button, circle pause or circle play button, and skip next button.", 

with the following paragraph:

Lines 112 to 116

“Furthermore, the track's duration in minutes and seconds is shown on the right side, along with a blue slider ribbon representing the current position along the track. Finally, three icons are displayed on the navigation footer to control track reproduction: the Skip Previous button (on the left), the circular Pause or Play button (in the center), and the Skip Next button (on the right).”

Regarding Figure 4, we added the following activities:

● “No interest in study”

● “Data analysis”

Furthermore, we update the following lines in the paper for each one of these activities, as follows:

● Lines 149 to 150: “Some of them may refuse to participate in the study and thus be excluded from it (fifth stage).”

● Lines 159 to 160: “Finally, the data obtained from the questionnaires and from using the audiobook itself will be analyzed (eleventh stage).”

Finally, we decided against describing the activities in the figure in further detail. We consider that the text in the “Design” and “Recruitment” sections describes Figure 3 in enough detail, and that adding more text there would be redundant and confusing to some readers. 

Comment 2. There was no list of references attached to this manuscript. There were errors in the presentation of the cited reference in the PDF version e.g The cited references in the text indicated [?}.

Response 2: We apologize for the inconvenience. Both the references and the in-text citations should have been included. We don't know if it was a system error or ours, but we take full responsibility for it.

Both the citations and the references are now included in the article.

Comment 3. Line 91, the statement ‘The goal of this protocol is to describe a quasi-experimental study’ to be revised.

Comment 4. Line 94-95, the statement ‘Furthermore, this study will evaluate the usability of the mobile application designed for this population. The duration of the study will be 4 weeks.’ is to be written in the form of an objective and in one sentence.

Response 3 and 4: The statements mentioned in Comments 3 and 4 have been revised. The following paragraph is more concise and consistent with the objective stated in the summary.

Lines 74 to 77: 

“The goal of this 4-week quasi-experimental study is to assess the impact of an audiobook mobile application on the well-being perception of older adults belonging to a Community Rehabilitation Center (CRC) at Concepcion, Chile, and to evaluate the usability of the mobile application designed for this population.”

Comment 5. More information on the data storage/server is to be provided.

Response 5. Thank you very much for your comment. We added the following paragraph in lines 128 to 132:

“The remote database will be stored on a paid server, which guarantees the privacy, accuracy, and availability of the stored data, as well as avoiding any damage or misuse by providing protection against unauthorized access and data leaks. Access to this database will be allowed only to one of the researchers (POR), who can download the information for analysis. Afterward, this information will be deleted from the server.”

Comment 6. Line 170-175, proper sample size calculation is to be provided, before and after the inclusion of attrition rates. Sample size formula or software used to derive the sample size is to be stated.

Response 6.

Given the limited research on audiobook use, we used the study by Ameri et al. [28], which examines an outcome related to mental health. This study follows a similar approach to ours but evaluates changes in the perceived well-being of older adults, encompassing social, hedonic, and eudaimonic well-being.

In addition, the scale used in [28], the standard mental health questionnaire (SCL-90-R), is comparable to the one employed in our study. Both use a Likert scale. Concerning the attrition rate, if some of the participants drop out of the study during the intervention (for different reasons), it will be necessary to recruit more participants to ensure a sample size of 60 individuals.

Finally, the software used to obtain the sample size was “Power and Sample Size Calculation,” version 3.1.6 (October 2018) by William D. Dupont and Walton D. Plummer, Jr. (https://biostat.app.vumc.org/wiki/Main/PowerSampleSize). A reference to this software [29] will be incorporated into the article (line 168). 

Finally, we changed the first paragraph of the “Recruitment” section for the following one:

Lines 165 to 171:

“The participants will be recruited from the Community Rehabilitation Center at Concepción, Chile. We estimated our sample size considering a pre-post difference of 1,24 points on the mental health outcome [28] and a potential standard deviation of 3,4. Thus, we planned for a sample size of 60 individuals [29] with a continuous response variable measured pre-post intervention to be able to reject the null hypothesis that the response difference is zero with power 0,8 and an error probability of 0,05. The sampling technique will be non-probabilistic to be used for the convenience of the study.”

28. Ameri F, Vazifeshenas N, Haghparast A. The Impact of Audio Book on the Elderly Mental Health. Basic and Clinical Neuroscience Journal. 2017;8(5):361–370. doi:10.18869/nirp.bcn.8.5.361.

29. Dupont WD, Plummer Jr WD. Power and Sample Size Calculation; 2018.

Version 3.1.6. Available from:

https://biostat.app.vumc.org/wiki/Main/PowerSampleSize.

Comment 7. In Page 7 Line 185, it was written Figure 4 Study Design but in the attachment is was written Figure 3.

Response 7. We apologize for the confusing name. The attachment corresponding to Figure 4 (Study design) was called “studydesign_3.eps”, where the suffix 3 refers to a correlative number. To avoid confusion, we have changed the filename to “Fig4.eps”, as indicated in the guidelines for authors.

Comment 8. (a) Quasi-experimental designs typically requires a comparison group. The reason to conduct one-arm design is to be stated. (b) Although some information about changes in the outcome over time within the same group could still be obtained, it is difficult to determine if the observed changes in the outcome are actually due to the intervention being studied. (c) Also, it is more susceptible to threats of internal validity.

Response 8 (a) The study presents a quasi-experimental pretest/posttest design with a single group, which implies comparing the same participants before and after the intervention. This single-arm design is justified for four main reasons [25-27]: (i) ethical considerations, since the aim is for the entire target population who meet the eligibility criteria to benefit from the intervention without facing the ethical dilemmas associated with the randomization of patients. It should be noted that an accredited Research Ethics Committee has approved the study. (ii) in a tight community, it is difficult to avoid contamination of the control group since participants tend to share information and technical devices when they meet. Thus, measures in the control group will bias the results to the null because they will be unintentionally exposed to the intervention. (iii) Based on the study by Ameri et al. [28], the difference in the pre-post interventional group was significant, so using a control group may not be necessary. (iv) According to our analysis, the use of a quasi-experimental design offers a more cost-effective and feasible approach for implementing this research.

According to above, we added the following paragraph in the line 136 to 139.

“This design is based on the desire to reduce ethical considerations (ethical problems of randomizing patients), avoid data contamination, and offer a more cost-effective and feasible approach for implementing this research [25-27].”

25. Handley MA, Lyles CR, McCulloch C, Cattamanchi A. Selecting and Improving Quasi-Experimental Designs in Effectiveness and Implementation Research.

Annual Review of Public Health. 2018;39:5–25. doi:10.1146/annurev-publhealth-040617-014128.

26. Harris AD, McGregor JC, Perencevich EN, Furuno JP, Zhu J, Peterson DE, et al. The Use and Interpretation of Quasi-Experimental Studies in Medical Informatics. Journal of the American Medical Informatics Association. 2006;13(1):16–23. doi:10.1197/jamia.M1749.

27. Maciejewski ML. Quasi-experimental design. Biostatistics & Epidemiology. 2020;4(1):38–47. doi:10.1080/24709360.2018.1477468.

28. Ameri F, Vazifeshenas N, Haghparast A. The Impact of Audio Book on the Elderly Mental Health. Basic and Clinical Neuroscience Journal. 2017;8(5):361–370. doi:10.18869/nirp.bcn.8.5.361.

Response 8 (b). We appreciate your comment. Thus, we considered analyzing confounding variables such as demographics, discussed on lines 295 to 313 in the “Statistical Analysis” section. 

At the same time, as is discussed in the “Recruitment” section, lines 196 to 198, in order to guarantee that the questionnaire allows us to know the participants' perception of well-being, both the content and face validity were made through an expert committee.

Response 8 (c). We agree with the comment. This is why, in this study, threats to internal validity were addressed through the study design, assessment tools, and statistical methods employed, as we have commented above.

Comment 9. Line 197, the information on the validity of DIGCOMP in the context of the local population is to be provided. Likewise with the System Usability Scale. Ensure all questionnaire validation information is described where applicable. The scoring method is to be provided if any.

Response 9. We concur with this comment. In particular, we have decided to change the digital competences questionnaire DIGCOMP [30] to a recently validated Spanish questionnaire based on DIGCOMP [31]. On the other hand, we added the reference to a Spanish version [34] of the SUS [33], with another reference on how to interpret its results [35].

According to the above, we have added the following sentences to the article:

• Line 193: “with a validated Spanish questionnaire based on DIGCOMP [30,31]”.

• Lines 222 to 223: “a validated Spanish usability questionnaire (SUS [33,34])”.

• Lines 241 to 242: “The way to categorize the digital competences of each participant is described in the same work by García Díaz and Villafañe [31]”.

• Lines 256 to 258: “The overall score calculation method is detailed in the work of Sevilla-Gonzalez et al. [34], and the results interpretation can be found in Bangor et al.'s research [35].”.

Also, we changed some sentences in the article by the following ones:

● From: “It will be measured through the DIGCOMP questionnaire [34]. In it, four areas of relevant competences are identified: information, communication, content creation and problem solving.”

To (lines 236 to 240):

“It will be measured through a questionnaire based on DIGCOMP [31]. In it, five areas of relevant competences are identified: information and data search, communication and collaboration, digital content creation, safety, and problem solving.”.

● From: “It consists of 10 questions that are assessed with a 3-point Likert subjective scale (1 totally disagree and 3 completely agree).”

To (lines 252 to 254):

“It consists of 10 questions that are assessed with a 5-point Likert subjective scale (1 strongly disagree and 5 strongly agree).”.

Finally, we have decided not to provide the scoring method for both DIGCOMP and SUS. This decision was based on the fact that, in our opinion, the description of the method for obtaining the final score for each questionnaire may be too lengthy and distract the reader. For this reason, references and comments have been included in the article so that the reader can refer to them if necessary.

30. Ferrari A. Digital Competence in Practice: An Analysis of Frameworks. Research Centre of the European Commission; 2012. EUR 25351 EN.

31. García Díaz F, Villafañe S. Habilidades digitales en la provincia de Córdoba. Comisión Económica para América Latina y el Caribe (CEPAL); 2024. Available from: https://hdl.handle.net/11362/80590.

33. Brooke J. SUS-A quick and dirty usability scale. Usability Evaluation in Industry. 1996;189(194):4–7.

34. Sevilla-Gonzalez MDR, Moreno Loaeza L, Lazaro-Carrera LS, Bourguet Ramirez B, Vázquez Rodríguez A, Peralta-Pedrero ML, et al. Spanish Version of the System Usability Scale for the Assessment of Electronic Tools: Development and Validation. JMIR Human Factors. 2020;7(4):e21161. doi:10.2196/21161.

35. Bangor A, Kortum PT, Miller JT. An Empirical Evaluation of the System Usability Scale. International Journal of Human–Computer Interaction. 2008;24(6):574–594. doi:10.1080/10447310802205776.

Comment 10. Description on missing data (if any) and handling method is to be provided.

Response 10. At the end of the “Statistical Analysis” section, we added a paragraph briefly describing how we will handle missing data, with the corresponding reference.

Lines 300 to 303:

“Data analysis will be performed under an intention-to-treat (ITT) approach by a researcher blinded to the study group. The multiple imputation technique [36] will be applied using the predictive mean matching method to handle missing data and ensure the inclusion of all participants in the analysis.”

36. Sterne JAC, White IR, Carlin JB, Spratt M, Royston P, Kenward MG,

---

## [Decision Letter · Decision Letter 1]

8 Oct 2024

How Does a Poetry Audiobook App Improve the Perception of Well-being in Older Adults? A Study Protocol

PONE-D-24-11113R1

Dear Dr. Rossel,

We’re pleased to inform you that your manuscript has been judged scientifically suitable for publication and will be formally accepted for publication once it meets all outstanding technical requirements.

Kind regards,

Junchen Shang

Academic Editor

PLOS ONE

Additional Editor Comments (optional):

Reviewers' comments:

Reviewer's Responses to Questions

**Comments to the Author**

1. Does the manuscript provide a valid rationale for the proposed study, with clearly identified and justified research questions?

Reviewer #1: Yes

2. Is the protocol technically sound and planned in a manner that will lead to a meaningful outcome and allow testing the stated hypotheses?

Reviewer #1: Partly

3. Is the methodology feasible and described in sufficient detail to allow the work to be replicable?

Reviewer #1: Yes

4. Have the authors described where all data underlying the findings will be made available when the study is complete?

Reviewer #1: Yes

5. Is the manuscript presented in an intelligible fashion and written in standard English?

Reviewer #1: Yes

6. Review Comments to the Author

You may also provide optional suggestions and comments to authors that they might find helpful in planning their study.

Reviewer #1: The authors have put in great effort to address the comments. The manuscript is now acceptable for publication.

7. PLOS authors have the option to publish the peer review history of their article (what does this mean?). If published, this will include your full peer review and any attached files.

Reviewer #1: No

---

## [Editor Report · Acceptance letter]

20 Oct 2024

PONE-D-24-11113R1 

PLOS ONE

Dear Dr. Rossel, 

I'm pleased to inform you that your manuscript has been deemed suitable for publication in PLOS ONE. Congratulations! Your manuscript is now being handed over to our production team.

Kind regards, 

on behalf of

Dr. Junchen Shang 

Academic Editor

PLOS ONE